# Vitamin D Nutritional Status of Chinese Pregnant Women, Comparing the Chinese National Nutrition Surveillance (CNHS) 2015–2017 with CNHS 2010–2012

**DOI:** 10.3390/nu13072237

**Published:** 2021-06-29

**Authors:** Yichun Hu, Rui Wang, Deqian Mao, Jing Chen, Min Li, Weidong Li, Yanhua Yang, Liyun Zhao, Jian Zhang, Jianhua Piao, Xiaoguang Yang, Lichen Yang

**Affiliations:** Key Laboratory of Trace Element Nutrition of National Health Commission, National Institute for Nutrition and Health, China CDC, Beijing 100050, China; huyc@ninh.chinacdc.cn (Y.H.); wangrui@ninh.chinacdc.cn (R.W.); maodq@ninh.chinacdc.cn (D.M.); chenjing@ninh.chinacdc.cn (J.C.); limin@ninh.chinacdc.cn (M.L.); liwd@ninh.chinacdc.cn (W.L.); yangyh@ninh.chinacdc.cn (Y.Y.); zhaoly@ninh.chinacdc.cn (L.Z.); zhangjian@ninh.chinacdc.cn (J.Z.); piaojh@163.com (J.P.); yangxg@ninh.chinacdc.cn (X.Y.)

**Keywords:** vitamin D, Chinese pregnant women, 25-hydroxyvitamin D, vitamin D supplements

## Abstract

Optimal vitamin D (vitD) status is beneficial for both pregnant women and their newborns. The aim of this study was to evaluate the vitamin D status of Chinese pregnant women in the latest China Nutrition and Health Surveillance (CNHS) 2015–2017, analyze the risk factors of vitamin D deficiency (VDD), and compare them with those in CNHS 2010–2012. Serum 25 hydroxyvitamin D (25(OH)D) was measured by ELISA method. City type, district, latitude, location, age, vitamin D supplements intake, education, marital status, annual family income, etc., were recorded. The median 25(OH)D concentration was 13.02 (10.17–17.01) ng/mL in 2015–2017, and 15.48 (11.89–20.09) ng/mL in 2010–2012. The vitamin D sufficient rate was only 12.57% in 2015–2017, comparing to 25.17% in 2010–2012. The risk factors of vitamin D inadequacy (25(OH)D < 20 ng/mL) in 2015–2017 were not exactly consistent with that in 2010–2012. The risk factors included season of spring (*p* < 0.0001) and winter (*p* < 0.001), subtropical (*p* < 0.001), median (*p* < 0.0001) and warm temperate zones (*p* < 0.0001), the western (*p* = 0.027) and the central areas (*p* = 0.041), while vitD supplements intake (*p* = 0.021) was a protective factor in pregnant women. In conclusion, vitD inadequacy is very common among Chinese pregnant women. We encourage pregnant women to take more effective sunlight and proper vitD supplements, especially for those from the subtropical, warm and medium temperate zones, the western and the central, and in the seasons of spring and winter.

## 1. Introduction

Vitamin D (calciferol, vitD) is an essential fat-soluble vitamin, and it is well-known that it is beneficial for bone health. With the increasing attention of vitD, researchers found that vitamin D deficiency (VDD) may be relative to cardiovascular diseases [1], type 2 diabetes [2], certain types of cancer [3], auto-immune diseases [4], depression [5] and autism [6]. VDD is very common all over the world and has been a major public health problem worldwide in all age groups. Among them, pregnant women are at high risk of VDD [7,8,9]. In recent years, many studies have extensively investigated the association between VDD during pregnancy and an increased risk of late pregnancy complication, such as bone impairment, osteoporosis, hypocalcaemia, gestational diabetes, preeclampsia, premature birth, hypertension, caesarean section, and bacterial vaginosis [8,10,11], together with various adverse neonatal outcomes, including poor fetal growth, small of gestational age infants, lower birth weight infant [12], rickets, infantile eczema and stunting in the first years of life [13]. Some recent studies have reported that the vitD status of mother during pregnancy would also affect the fetal bone and immune development [11].

25-hydroxyvitamin-D (25(OH)D) is the major circulating form of vitD in the blood, and it was a reliable biological indicator for vitD status assessment [14]. The 25(OH)D concentration is constant throughout pregnancy [15] and the mother is the only source of vitD for the fetus [15,16]. The 25(OH)D concentration in cord blood is about 60–89% of maternal concentration in blood [15]. In all, maintaining optimum vitD nutrition during pregnancy is essential for the prevention of hypovitaminosis D in the fetus and VDD at birth and in early infancy.

Our group pays great attention to the nutritional status of vitD in the population, especially for pregnant women. We have published the result of the fifth Chinese national nutrition survey, namely China Nutrition and Health Surveillance (CNHS) 2010–2012, and only 25.2% of pregnant women have adequate levels of vitD [17]. We also participate in the latest CNHS 2015–2017. The aim of this study was to evaluate the vitD status of pregnant women in the CNHS 2015–2017, and analyze the risk factors for VDD. Moreover, we also compared the vitD status and risk factors of pregnant women in CNHS 2015–2017 with that in CNHS 2010–2012.

## 2. Materials and Methods

### 2.1. Subjects and Ethics

The data of this study was obtained from the fifth and the sixth round of the Chinese national nutrition surveillances, CNHS 2010–2012 and CNHS 2015–2017, respectively. They were cross-sectional surveys of the civilian non-institutionalized population of China and conducted by the Chinese Center for Disease and Control Center (China CDC) as well as the past 4 rounds of national nutrition surveillance conducted in 1959, 1982, 1992, 2002 respectively [18]. All the participants in the latest two rounds of survey were selected by using a stratified, multistage, and probability-based random sampling scheme. All county (district) level administrative units (including counties, county-level cities and districts) from 31 provinces of Chinese mainland were covered. People with serious physical and mental diseases were excluded. 30 Pregnant women were selected from each county-level maternal and child health care institution where the surveillance site located. There were 150 county-level surveillance sites in CNHS 2010–2012 [19] and 302 survey sites from CNHS 2015–2017 [20], respectively. The detailed sampling methods were described by the workbook of each round [20,21] and Yu et al. [22]. A total of 2250 serum samples of pregnant women in CNHS 2010–2012 were selected by stratified random sampling. 9060 pregnant women in CNHS 2015–2017 were investigated and had blood collected. All participants were given informed consent in writing to participate in the study.

### 2.2. Data Collection

National workgroup was established in China CDC to conduct the surveillance. The interview, anthropometric measurements and blood taken were carried at the scene. Age and nationality were confirmed by identity cards. Based on self-report, education, pregnancy frequency, gestation age, marital status, use of vitD supplements was recorded. All the information was recorded and logged into the systematic platform of national survey of nutrition and health status for Chinese residents [8]. Latitude retrieved from Baidu map and divided into tropical (0–23.5° N), subtropical (23.5–32.0° N), warm temperate (32.0–40.5° N) and medium temperate zones (40.5–46.5° N). Season was recorded according to the month of blood taken, spring (March to May), summer (June to August), autumn (September to November) and winter (December to February). China’s Qinling Mountains and Huaihe River were recognized as the boundary to divide the location into the Northern and the Southern. The city type was divided into urban and rural areas [19,20], and the district was divided into eastern, central and western [23]. The pre-pregnancy BMI level was classified as thin, normal, overweight and obesity, half a year before pregnancy [24]. The education level is divided into primary (primary school and below), medium (junior high school/high school/secondary school) and advanced (junior college or above); Marriage status is divided into married and unmarried (unmarried/divorce/widows). The annual family income per capita was divided into low, middle and high according to the three-digit method. The cutoffs of annual family income per capita were RMB 10,000 and RMB 25,000 in CHNS 2010–2012, and it were RMB 13,333.33 and RMB 28,000 in CNHS 2015–2017.

### 2.3. Blood Sample and 25-Hydroxyvitamin D Measurement

The anthropometric measurements adopted unified equipment and methods. The fasting venous blood from each pregnant woman was collected, standing for 30 min and then centrifuged at 1500× *g* for 15 min. The serum was separated, divided and stored in brown vessel at −20 °C in the laboratory where the surveillance site was located. All the samples from each the surveillance site were transported to the National Institute for Nutrition and Health, China CDC by cold chain and then stored at −70 °C refrigerator before measurements. Serum 25(OH)D concentration were measured in our laboratory by the ELISA method on both rounds of the surveillance (Immuno Diagnostic System Ltd., Boldon, UK). 10% serum samples were used for double sample determination. The interassay coefficients of variation (CVs) of quality control samples in CNHS 2010–2012 were 3.8% and 3.5% at 9.9 ng/mL and 23.0 ng/mL, respectively. The CVs of quality control samples in CNHS 2015–2017 were 6.7% and 6.3% at 12.8 ng/mL and 43.2 ng/mL, respectively.

A serum 25(OH)D concentration of a greater than or equal to 20 ng/mL is considered sufficient. The serum 25(OH)D concentration which was lower than 20 ng/mL but higher or equal to 12 ng/mL was considered insufficient, and if it was lower than 12 ng/mL, it was considered deficient [25]. The vitD deficiency and insufficiency were grouped together as vitD inadequacy (25(OH)D < 20 ng/mL) in this study.

### 2.4. Data Analyses

All the data were analyzed by SAS 9.4 software (SAS Institute, Cary, NC, USA). According to different hypothesized predictors, all the participants in this study were divided into different sub-groups for vitD nutritional status as described in 2.2. Serum 25(OH)D concentration was record by P50 (P25~P75) due to inconsistent with the normal distribution by the normality test, and then they were compared by Kruskal–Wallis test. Frequencies were presented as percentages (%) and the prevalence rates of subgroups were compared by chi square test. The univariable and multivariable logistic regression analysis were utilized to analyze the relationship between vitD inadequacy and possible predictors (e.g., age, city type, latitude, nationality, gestational age, pregnant frequency, season, pre-pregnancy BMI, vitD supplement intake, education, marital status, and annual family income per capita). Considering the overlap of latitude and location, we deleted the location variable in the logistic regression analysis. The odds ratio (OR) and 95% confidence intervals (CIs) were determined by multivariable logistic regression models. The difference was statistically significant with *p* < 0.05.

## 3. Results

### 3.1. Serum 25 Hydroxyvitamin D Concentration in CNHS 2015–2017 and CNHS 2010–2012

The serum 25(OH)D concentration of 2006 pregnant women in CNHS 2010–2012 and 8200 pregnant women in CNHS 2015–2017 were showed in Table 1, after excluding those unqualified samples such as hemolysis or insufficient serum volume, and/or with incomplete data.

In the CNHS 2015–2017, the median age of pregnant women was 27.4 years old (interquartile range (IQR) 25.0–30.4 y), 92.12% of them were below 35 y. 59.87% of participants were from urban areas, and the Han nationality accounted for 88.06% of the participants. 41.76% pregnant women were in the second trimester. The median BMI was 23.94 kg/m^2^, and 46.71% of them were normal in BMI before pregnancy. Only 0.37% blood samples were taken from summer, while 65.90% were from winter. 4.48% pregnant women claimed to have taken vitD supplements. 41.77% of participants had education of college degree and above, and 0.82% pregnant women were unmarried.

In the CNHS 2010–2012, the median age of pregnant women was 26.1 years old (IQR 23.6–29.2), 94.47% was below 35 y. 51.20% of participants were from urban areas. 63.71% of participants were pregnant for the first time. The median BMI before pregnancy was 23.89 kg/m^2^. 46.81% of participants were normal in BMI before pregnancy. Only 5.23% blood samples were taken from spring and summer. 8.97% pregnant women claimed to have taken vitD supplements. 60.87% of participants had junior high school, high school or technical secondary school and 1.89% of participants were unmarried.

The serum median 25(OH)D concentration in CNHS 2015–2017 was 13.02 (10.17–17.01) ng/mL, and it was 15.48 (11.89–20.09) ng/mL in CNHS 2010–2012 (Table 1). Compared with the CNHS 2010–2012, the serum 25(OH)D concentration in CNHS 2015–2017 decreased significantly. Except for summer in the seasonal subgroups and unmarried in marital subgroups, the levels of serum 25(OH)D in all the other subgroups in CNHS 2015–2017 were significantly decreased than in CNHS 2010–2012 at different degrees.

Pregnant women aged 18–24.9 y had the lowest 25(OH)D concentration in CNHS 2010–2012, while there was no difference between different age groups in CNHS 2015–2017. The blood taken in summer and autumn had higher 25(OH)D concentration and then followed by winter and spring in CNHS 2010–2012, while it was highest in summer and then followed by autumn, winter and spring in CNHS 2015–2017. It was shown that the pregnant women taking vitD supplements and pregnant women with medium education level had a higher 25(OH)D concentration in CNHS 2015–2017. However, no such trend was found in the CNHS 2010–2012.

In both of the two rounds nutrition surveillance, the pregnant women from eastern areas were both significantly higher than those from the central and the western areas. Southern pregnant women had higher 25(OH)D concentration than those from the North. With the increase of latitude, the level of serum 25(OH)D decreased gradually. The pregnant women from tropical zone had the highest serum 25(OH)D concentration and then followed by the subtropical zone. The 25(OH)D concentration of Han pregnant women is higher than that of ethnic minorities. Pregnant women who were pregnant for the first time had lower 25(OH)D concentration. Pregnant women with lower pre-pregnant BMI had higher concentration of serum 25(OH)D. No differences were found in different city type, gestational age, marital status, annual family income per capita in terms of 25(OH)D concentration.

### 3.2. Vitamin D Status of Pregnant Women in CNHS 2015–2015 and CNHS 2010–2012

The vitD sufficient rate of pregnant women in CNHS 2015–2017 was less than a half that in CNHS 2010–2012. Except for summer in seasonal subgroups and unmarried in marital status’ subgroups, the sufficient rate in all the other subgroups in CNHS 2015–2017 decreased significantly than that of CNHS 2010–2012.

As shown in Table 2, in 2015–2017, only 12.57% pregnant women had sufficient vitD, 45.46% were insufficient and 41.96% were deficient. The pregnant women from tropical areas had highest vitD sufficiency that was up to 41.42%, while the lowest sufficiency was those from the northern areas as low as 4.49%. The lowest VDD of pregnant women was found in summer as low as 3.33%, while the highest VDD was found in pregnant women lived in the warm temperate zones and it was 59.03%.

In 2010–2012, 25.17% pregnant women were sufficient in vitD, 49.30% were insufficient and 25.52% were deficiency. The pregnant women lived in tropical areas had highest vitD sufficiency which was up to 59.84% and the lowest VDD rate as low as 3.28%. The unmarried pregnant women had the lowest sufficiency, 10.53%.

In both rounds of CNHS, pregnant women live in the tropical areas had the lowest VDD prevalence among all the subgroups. The VDD rate of pregnant women from eastern areas was significant lower than those from the central areas and the western areas. The prevalence of VDD in pregnant women from the northern areas were about twice that from the southern areas. The VDD rate in summer was lowest and it was highest in spring. Thin pregnant women had a statistical lower VDD rate.

The Han nationalities had a significant lower vitD deficiency than those of minorities in CNHS 2010–2012, but the difference did no longer exist in CNHS 2015–2017. Comparing to CNHS 2010–2012, there were significant differences in VDD prevalence among different city types, pregnancy frequency, vitD supplement intake, education level, marital status and annual family income per capita in CNHS 2015–2017. Pregnant women lived in urban areas had statistical higher VDD rate than that in rural areas. Pregnant women who were taking vitD supplements had statistical lower VDD prevalence than those not taking supplements. And those with medium education level had lower VDD prevalence. Pregnant women who had high annual family income per capita had lower VDD prevalence. It is interesting that married pregnant women had higher prevalence of vitD sufficiency in CNHS 2010–2012, while in turn married pregnant women had higher VDD in CNHS 2015–2017. The change of vitD insufficiency during the two rounds of survey was not as much as that of deficiency rate and sufficiency rate.

### 3.3. Risk Factors for Vitamin D Insufficiency and Deficiency

In 2010–2012, the result of univariable logistic regression analysis showed that the vitD inadequacy was stronger associated with western [odds ratio (OR) = 2.764, *p* < 0.0001, relative to eastern], warm temperate zones (OR = 8.553, *p* < 0.0001, relative to tropical zones), ethnic minorities (OR = 1.521, *p* = 0.029, relative to Han nationality), overweight and obesity (OR = 1.189 *p* = 0.041, for obesity, and OR = 1.318, *p* < 0.001, for overweight, relative to normal bodyweight), unmarried (OR = 2.902, *p* = 0.045 relative to married). Pregnant women aged 30.0–34.9 y (OR = 0.611, *p* = 0.024, relative to 18–24.9 y) and the pregnant women whose pre-pregnancy BMI were lower than 24.0 were less likely to suffer from vitD inadequacy (OR = 0.462, *p* < 0.0001, relative to normal bodyweight). On the basis of univariable logistic regression, the district, latitude, age group, nationality, season, pre-pregnancy BMI, and marital status were brought to the multivariable logistic regression model to further screening and refining effective indicators. The results showed that risk factors for vitD inadequacy included: pregnant women from western areas (OR = 3.746, *p* < 0.001, relative to eastern areas), warm temperate zones (OR = 9.388, *p* < 0.0001, relative to tropical zones), and in autumn and winter (OR = 9.251, *p* = 0.005, for autumn; OR = 14.055, *p* < 0.0001, for winter, relative to whose blood was taken in summer), overweight (OR = 1.424, *p* = 0.038, relative to normal bodyweight) and who were unmarried (OR = 9.715, *p* = 0.006, relative to married). 30.0~34.9 y in age group would reduce the risk of vitD inadequacy (OR = 0.661, *p* = 0.048, relative to 18–24.9 y) (Figure 1).

In 2015–2017, the vitD inadequacy was strongly associated with the central and the western areas (OR = 2.500, *p* < 0.0001, for the central; OR = 2.318, *p* < 0.0001, for the western; relative to the eastern), ethnic minorities (OR = 1.269, *p* = 0.031, relative to Han nationality), subtropical, warm temperate and medium temperate zones (OR: 3.287, 13.099, 13.412 respectively, *p* < 0.0001, relative to the tropical zones), spring and winter (OR = 4.037, *p* < 0.0001 for spring; OR = 3.284, *p* = 0.0013 for winter, relative to summer), overweight and obesity (OR = 1.168, *p* = 0.010 for overweight; OR = 1.243, *p* = 0.003 for obesity, relative to normal bodyweight), not taking vitD supplements (OR = 1.622, *p* = 0.0005, relative to taking vitD supplements) and low annual family income per capita (OR = 1.677, *p* < 0.0001, relative to high income level). Pregnant woman who were thin before pregnancy (OR = 0.668, *p* < 0.001, relative to normal bodyweight) and who were with the third or more pregnancies (OR = 0.730, *p* = 0.001, relative to pregnant with the first child) had lower risk of vitD inadequacy. And then the result of multivariable logistic regression showed that pregnant woman who were from the central and the western areas (OR = 2.080, *p* = 0.041, for the central; OR = 2.121, *p* = 0.027, for the western; relative to the eastern), subtropical, warm temperate and medium temperate zones (OR = 2.818, *p* < 0.001, for the subtropical zones; OR = 12.275, *p* < 0.0001, for the warm temperate zones; OR = 11.972, *p* < 0.0001, for the medium temperate zones; relative to the tropical zones), and whose blood was taken in spring and winter (OR = 10.007, *p* < 0.0001, for spring; OR = 5.883, *p* < 0.001, for winter; relative to whose blood was taken in summer) had increased risks to have vitD inadequacy. Taking vitD supplements reduced the risk of vitD inadequacy (OR = 0.683, *p* = 0.021, relative to those not taking supplements) (Figure 1).

In both rounds of the national nutrition survey, the pregnant women from the western areas, warm temperate zones, and whose blood was taken in winter were risk factors for vitD deficiency and insufficiency. Comparing to that of CNHS 2010–2012, the unmarried, overweight and northern were no longer risk factors for pregnant women, and 30.0~34.9 y was no longer protective factor either. Yet, vitD supplements intake has become a protective factor for vitD inadequacy in CNHS 2015–2017.

## 4. Discussion

VDD is widespread all around the word and has been a public health problem in all age groups [26]. The older person, pregnant women and non-western immigrants are the most risk groups for VDD, on a global scale [7]. As a special group, the nutritional status of pregnant women is not only related to their own health, but also directly affects the growth and development of the fetus. All over the world, the prevalence of vitD insufficiency/deficiency was ranging from 18% to 98%, depending on the location of country, local clothing customs and lifestyle [27,28]. In low UV exposure area, the deficiency rate (serum 25(OH)D < 20 ng/mL) as high as 98% [11]. Recognition that VDD is a worldwide public health problem in pregnant women is the basis to modify public health strategies to improve the vitD nutrition of mother and child, and reduce the burden of diseases [29].

The median concentration of serum 25(OH)D in Chinese pregnant women was significantly lower than that of all the other age group in both CNHS 2010–2012 [17,30] and CNHS 2015–2017 [Not published]. The median 25(OH)D concentration in pregnant women in both rounds of CNHS was below the global goal of VDD prevention, 20 ng/mL all year long in all the population [7]. The vitD insufficiency and deficiency (serum 25(OH)D < 20 ng/mL) of Chinese pregnant women in 2010–2012 was 74.82% and increased to 87.42% in 2015–2017, showing an upward trend. The median of 25(OH)D for pregnant women in 2015–2017 was less than half of the standard that 30 ng/mL as “sufficient” for pregnant and lactating women and young children recommended by the Canadian Pediatric Society (CPS) [31]. Thus the VDD of pregnant women in China has been already in a serious state, great attention must be paid.

In 2010–2012, the most risk factor for vitD inadequacy was season of winter, and then followed by season of autumn and unmarried marital status. The season of spring, warm and medium temperate zones were the most three risk factors of VDD in 2015–2017. Due to the actual situation of monitoring implementation, there are limited samples distributed in summer. However, the results of this study still reflect the influence of season on 25(OH)D concentration in blood. The seasonal variation trend for vitD level and vitD inadequacy are consistent with Chao et al. reported. The vitD level of the Asian population is generally higher in summer than in winter [31]. The effects of seasonal variations on vitD deficiency and insufficiency were also widely reported in both pregnant women and the other population [11,32]. In addition to the sunshine exposure time, the severe haze of China may also be the cause of insufficient sun exposure [33]. In our study, the 25(OH)D concentration of warm temperate were lowest than the other climatic zones, and it was a risk factors for VDD in both rounds of survey. This was similar with many researchers have reported that the level of vitD was decreased with higher geographical latitude [15,34] because of less ultraviolet (UV) B radiation. The relationship between marital status and VDD in pregnant women showed an opposite trend. This may be related to changes in multiple factors such as lifestyle, environment, etc. The western areas were always the risk factors for the Chinese pregnant women in both rounds of CNHS. We cannot find similar trends reported elsewhere and this might be related to the changes in the environment. Further study should be made to reveal the deep causes.

As time went by, some risk factors of vitD inadequacy in pregnant women also changed. Marital status and BMI are no longer risk factors for vitD inadequacy in CNHS 2015–2017. 80% or more vitD of our requirement is mainly produced in response to the exposure of skin to sunlight through UV-B synthesis [35,36]. However, it’s very difficult for women to get effective and sufficient UV radiation, because they are more likely to stay indoors or shade, use sunscreen, covers the skin with clothes [36,37], coupled with the increasingly severe haze weather in recent years. Due to the changes in lifestyle and environment, the differences in age, BMI and marital status existing in the past are no longer apparent in CNHS 2015–2017 in terms of risk factors for vitD inadequacy in pregnant women.

In our study, although only 4.48% pregnant women claimed to have taken vitD supplements, the average of 25(OH)D concentration in different gestational periods was higher than that of pregnant women who did not take vitD supplements. No statistical difference between three gestational periods. The result of 25(OH)D concentration, rate of vitD sufficient prevalence and logistic regression still showed that it has been a significant protective factor for pregnant women in the latest CNHS 2015–2017. Many studies have also proved the positive effect of supplements intake on the concentration of vitD in pregnant women [15]. Moreover, a Cochrane review found evidence that vitD supplements during pregnancy reduced the risk of preeclampsia, premature birth and low birth weight [38]. Roth et al. made a systematic review to suggest vitD supplements may have the potential benefits on a reduced incidence of preeclampsia, gestational diabetes, preterm labour, caesarean section and infection in mothers [15,39,40], an increase of birth weight, a reduced risk of small-for-gestational age (SGA), increase in length at 1 year of age, and a reduced risk of offspring asthma or recurrent/persistent wheeze up to 3 years of age in fetuses and infants [10,41]. Therefore, it is beneficial for both mother and fetus by taking vitD supplements to improve the vitD level on the basis of dietary reference intakes recommended by China Nutrition Society [42].

Our group reported serum 25(OH)D concentration for the first time in CNHS 2010–2012 [17,28]. The number of monitoring sites doubled, and the sample size has increased significantly in CNHS 2015–2017 with a national representative. The vitD status has been paid great attention and became the official monitoring item in CNHS 2015–2017 under our insistence. We reported the vitD status of pregnant women in the latest Chinese national nutrition survey. We understand that the bias caused by different detection methods, therefore we adopted the same method and reagents to detect the 25(OH)D concentration. We adopted the same method and reagents to detect the 25(OH)D concentration. On the basis, we conducted the comparison of vitD status in two rounds of surveillance among pregnant women for the first time.

We also acknowledge several limitations in this study. We were unable to assess the duration of sun exposure and the dietary sources of vitD and calcium intake levels for each participant. Basic information of self-report such as vitD supplements intake, pregnancy frequency, education, etc., might involve reminder or wishful thinking bias. Besides, due to the actual implementation of the field work, the samples originated from spring and summer in CNHS 2010–2012 and from summer in CNHS 2015–2017 were limited. The above situations will lead to bias.

## 5. Conclusions

In conclusion, vitD deficiency and insufficiency were very common at above 80% prevalence amongst pregnant women in China from the latest surveillance. Also, this is worse than it was five years ago. VitD supplements intake was an effective way to decrease the probability of vitD deficiency and insufficiency. Therefore, we encourage that pregnant women should take more outdoor activities to get enough and effective sunlight. Also, vitD supplements intake is encouraged in all pregnant women under the recommendation of dietary reference intakes, especially for those from the subtropical, warm and medium temperate zones, the western and the central, and in the seasons of spring and winter.

## Figures and Tables

**Figure 1 nutrients-13-02237-f001:**
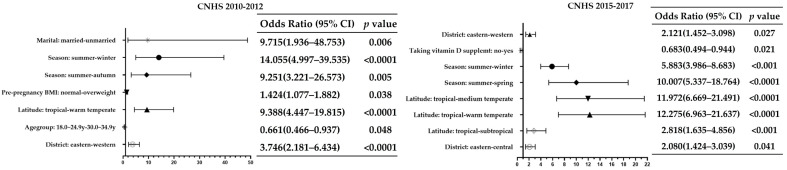
Risk factors for vitamin D inadequacy in CNHS 2010–2012 and CNHS 2015–2017 (vitamin D insufficiency and deficiency were grouped together as vitamin D inadequacy, 25(OH)D < 20 ng/mL).

**Table 1 nutrients-13-02237-t001:** Serum 25 hydroxyvitamin D concentration of pregnant women in CNHS 2010–2012 and CNHS 2015–2017 (P50 (P25~P75), ng/mL).

Characteristics	CNHS 2010–2012		CNHS 2015–2017		*p*
	N (%)	Concentration	N (%)	Concentration	
Total	2006	15.48 (11.89–20.09)	8200	13.02 (10.17–17.01)	<0.0001
City type					
Urban	1027 (51.20%)	15.41 (11.79–20.23)	4909 (59.87%)	12.94 (10.11–16.95)	<0.0001
Rural	979 (48.80%)	15.55 (11.95–19.91)	3291 (40.13%)	13.17 (10.26–17.11)	<0.0001
District					
Eastern	773 (38.53%)	16.89 (13.06–22.38) ^a^	2869 (34.99%)	14.19 (10.88–18.63) ^a^	<0.0001
Central	716 (35.69%)	15.55 (12.11–19.83) ^b^	2557 (31.18%)	12.44 (9.99–15.73) ^b^	<0.0001
Western	517 (25.77%)	13.74 (9.8–17.73) ^c^	2774 (33.83%)	12.65 (9.58–16.45) ^b^	0.003
Location					
Southern	1007 (50.20%)	17.11 (13.23–21.53) ^a^	3856 (47.02%)	15.05 (11.65–18.98) ^a^	<0.0001
Northern	999 (49.80%)	14.03 (10.91–18.34) ^b^	4344 (52.98%)	11.37 (9.11–14.18) ^b^	<0.0001
Age group					
18–24.9 y	785 (39.13%)	14.92 (11.46–19.13) ^b^	2055 (25.06%)	13.02 (10.13–16.95)	<0.0001
25–29.9 y	797 (39.73%)	15.85 (12.36–20.29) ^a^	3889 (47.43%)	12.92 (10.17–16.85)	<0.0001
30–34.9 y	313 (15.60%)	16.01 (12.16–21.53) ^a^	1610 (19.63%)	13.26 (10.19–17.36)	<0.0001
35 y+	111 (5.53%)	15.77 (12.06–20.3) ^ab^	646 (7.88%)	13.24 (10.44–17.19)	<0.001
Latitude					
tropical	130 (6.48%)	21.31 (16.37–24.98) ^a^	577 (7.04%)	18.63 (14.54–23.38) ^a^	<0.0001
subtropical	786 (39.18%)	16.65 (12.84–20.67) ^b^	3203 (39.06%)	14.98 (11.99–18.63) ^b^	<0.0001
warm temperate	776 (38.68%)	13.4 (10.68–17.59) ^c^	3202 (39.05%)	11.24 (9.00–14.18) ^c^	<0.0001
medium temperate	314 (15.65%)	16.19 (12.6–21.27) ^b^	1218 (14.85%)	11.83 (9.54–14.66) ^c^	<0.0001
Nationality					
Han	1813 (90.38%)	15.62 (12.06–20.29) ^a^	7221 (88.06%)	13.02 (10.23–17.09) ^a^	<0.0001
Ethnic minorities	193 (9.62%)	14.24 (10.29–18.45) ^b^	979 (11.94%)	12.9 (9.62–16.65) ^b^	<0.0001
Gestational age					
First trimester	462 (23.03%)	15.92 (12.6–20.53)	2352 (28.68%)	12.95 (10.22–16.74)	<0.0001
Second trimester	909 (45.31%)	15.59 (11.75–20.27)	3424 (41.76%)	13.09 (10.23–17.15)	<0.0001
Third trimester	635 (31.66%)	14.79 (11.8–19.12)	2424 (29.56%)	13.02 (9.98–17.11)	<0.0001
Pregnant frequency					
1	1278 (63.71%)	15.15 (11.73–19.42) ^b^	3739 (45.60%)	12.83 (9.97–16.69) ^b^	<0.0001
2	567 (28.27%)	15.79 (12.29–21.39) ^a^	3131 (38.18%)	13.16 (10.31–17.18) ^a^	<0.0001
3+	161 (8.03%)	16.79 (12.56–20.67) ^a^	1330 (16.22%)	13.42 (10.49–17.55) ^a^	<0.0001
BMI before pregnancy					
Thin	98 (4.89%)	18.8 (13.73–24.23) ^a^	305 (3.72%)	14.18 (10.88–18.44) ^a^	<0.0001
Normal	939 (46.81%)	15.69 (11.82–20.32) ^b^	3830 (46.71%)	13.29 (10.42–17.33) ^a^	<0.0001
Overweight	707 (35.24%)	14.98 (11.79–19.04) ^b^	2692 (32.83%)	12.88 (9.97–16.68) ^b^	<0.0001
Obesity	262 (13.06%)	15.62 (11.9–19.14) ^b^	1373 (16.74%)	12.68 (9.71–16.43) ^b^	<0.0001
Season					
Spring	55 (2.74%)	9.77 (8.46–11.22) ^c^	896 (10.93%)	12.27 (9.45–16.41) ^c^	<0.0001
Summer	50 (2.49%)	17.51 (13.29–23.08) ^a^	30 (0.37%)	17.52 (13.96–21.16) ^a^	0.968
Autumn	963 (48.01%)	16.76 (12.74–21.55) ^a^	1870 (22.80%)	14.76 (11.67–18.42) ^b^	<0.0001
Winter	938 (46.76%)	14.68 (11.43–18.95) ^b^	5404 (65.90%)	12.57 (9.84–16.42) ^c^	<0.0001
Vitamin D supplement intake					
No	1793 (89.38%)	15.47 (11.85–19.97)	7833 (95.52%)	12.96 (10.14–16.91) ^b^	<0.0001
Yes	180 (8.97%)	15.55 (12.73–21.29)	367 (4.48%)	14.89 (11.2–18.37) ^a^	0.003
unknown	33 (1.65%)	14.66 (10.56–19.52)	/	/	
Education					
Primary	145 (7.23%)	15.71 (11.32–20.77)	532 (6.49%)	12.63 (9.72–16.33) ^b^	<0.0001
Medium	1221 (60.87%)	15.55 (12.08–19.98)	4243 (51.74%)	13.3 (10.31–17.24) ^a^	<0.0001
Advanced	640 (31.90%)	15.35 (11.65–19.85)	3425 (41.77%)	12.8 (10.07–16.78) ^b^	<0.0001
Marital status					
Unmarried	38 (1.89%)	13.46 (11.79–17.74)	67 (0.82%)	14.57 (11.63–18.29)	0.492
Married	1968 (98.11%)	15.51 (11.9–20.16)	8133 (99.18%)	13.01 (10.16–16.98)	<0.0001
Annual family income per capita					
Low	630 (31.41%)	15.3 (11.83–19.37)	1734 (21.15%)	12.39 (9.56–15.94)	<0.0001
Mid	744 (37.09%)	15.25 (12.04–19.96)	1677 (20.45%)	12.64 (9.82–16.56)	<0.0001
High	462 (23.03%)	16.46 (11.89–21.00)	1890 (23.05%)	13.64 (10.63–17.73)	<0.0001
NR *	170 (8.47%)	15.73 (11.86–20.42)	2899 (35.35%)	13.35 (10.41–17.49)	<0.0001

^a,b,c^ refers to statistical difference in groups, *p* < 0.05; * NR—not reported.

**Table 2 nutrients-13-02237-t002:** Comparison of vitamin D status of pregnant women in CNHS 2010–2012 and CNHS 2015–2017 (%, 95% CI).

Characteristics	Sufficiency (25(OH)D ≥ 20 ng/mL)			Insufficiency (20 ng/mL > 25(OH)D ≥ 12 ng/mL)			Deficiency (25(OH)D < 12 ng/mL)		
	2010–2012	2015–2017	*p* Value	2010–2012	2015–2017	*p* Value	2010–2012	2015–2017	*p* Value
Total	25.17 (23.27–27.08)	12.57 (11.86–13.29)	<0.0001	49.30 (47.11–51.49)	45.46 (44.39–46.54)	0.002	25.52 (23.61–27.43)	41.96 (40.9–43.03)	<0.0001
City type									
Urban	25.71 (23.03–28.38)	12.43 (11.5–13.35)	<0.0001	48.30 (45.24–51.35)	44.59 (43.2–45.98)	0.031	26.00 (23.31–28.68)	42.98 (41.6–44.37)	<0.0001
Rural	24.62 (21.92–27.32)	12.79 (11.65–13.93)	<0.0001	50.36 (47.22–53.49)	46.76 (45.06–48.47)	0.048	25.03 (22.31–27.74)	40.44 (38.77–42.12) ^a^	<0.0001
District									
Eastern	32.60 (29.29–35.91) ^a^	19.21 (17.76–20.65) ^a^	<0.0001	49.42 (45.89–52.95)	46.36 (44.53–48.18)	0.130	17.98 (15.27–20.69) ^c^	34.44 (32.7–36.18) ^b^	<0.0001
Central	24.58 (21.42–27.74) ^b^	8.68 (7.59–9.77) ^b^	<0.0001	51.82 (48.15–55.48)	46.30 (44.37–48.24)	0.009	23.60 (20.49–26.72) ^b^	45.01 (43.09–46.94) ^a^	<0.0001
Western	14.89 (11.82–17.97) ^c^	9.30 (8.22–10.38) ^b^	<0.001	45.65 (41.35–49.95)	43.76 (41.92–45.61)	0.428	39.46 (35.24–43.68) ^a^	46.94 (45.08–48.79) ^a^	0.002
Location									
Northern	17.72 (15.35–20.09) ^b^	4.49 (3.83–5.14) ^b^	<0.0001	48.65 (45.55–51.75)	37.50 (35.97–39.03) ^b^	<0.0001	33.63 (30.7–36.57) ^a^	58.01 (56.46–59.57) ^a^	<0.0001
Southern	32.57 (29.68–35.47) ^a^	19.75 (18.57–20.94) ^a^	<0.0001	49.95 (46.86–53.04)	52.53 (51.05–54.02) ^a^	0.140	17.48 (15.13–19.83) ^b^	27.72 (26.39–29.05) ^b^	<0.0001
Age group									
18–24.9 y	21.53 (18.65–24.41) ^a^	11.97 (10.57–13.37)	<0.0001	49.68 (46.18–53.18)	45.40 (43.25–47.55)	0.041	28.79 (25.62–31.96)	42.63 (40.49–44.77)	<0.0001
25–29.9 y	26.47 (23.41–29.54) ^a,b^	12.24 (11.21–13.27)	<0.0001	50.19 (46.71–53.66)	45.41 (43.85–46.98)	0.014	23.34 (20.4–26.28)	42.35 (40.8–43.9)	<0.0001
30–34.9 y	30.99 (25.86–36.12) ^b^	13.91 (12.22–15.6)	<0.0001	45.37 (39.85–50.89)	45.22 (42.79–47.65)	0.961	23.64 (18.93–28.35)	40.87 (38.47–43.27)	<0.0001
35 y+	25.23 (17.14–33.31) ^a,b^	13.16 (10.55–15.77)	0.003	51.35 (42.05–60.66)	46.59 (42.75–50.44)	0.689	23.42 (15.54–31.31)	40.25 (36.47–44.03)	0.010
Latitude									
tropical	59.84 (51.13–68.54) ^a^	41.42 (37.40–45.44) ^a^	<0.001	36.89 (28.32–45.45) ^b^	41.59 (37.57–45.62) ^b^	0.407	3.28 (0.12–6.44) ^c^	16.98 (13.92–20.05) ^d^	<0.0001
subtropical	28.55 (25.4–31.71) ^b^	17.70 (16.38–19.02) ^b^	<0.0001	51.90 (48.41–55.4) ^a^	56.82 (55.11–58.54) ^a^	0.013	19.54 (16.77–22.31) ^b^	25.48 (23.97–26.99) ^c^	<0.001
warm temperate	14.83 (12.34–17.33) ^c^	5.12 (4.36–5.89) ^c^	<0.0001	48.72 (45.22–52.23) ^a,b^	35.85 (34.19–37.51) ^c^	<0.0001	36.45 (33.07–39.82) ^a^	59.03 (57.32–60.73) ^a^	<0.0001
medium temperate	28.98 (23.96–34) ^b^	5.01 (3.78–6.23) ^c^	<0.0001	49.04 (43.51–54.58) ^a,b^	42.69 (39.91–45.47) ^b^	0.043	21.97 (17.39–26.56) ^b^	52.30 (49.49–55.1) ^b^	<0.0001
Nationality									
Han	25.87 (23.85–27.89) ^a^	12.87 (12.09–13.64) ^a^	<0.0001	49.64 (47.34–51.94)	45.49 (44.34–46.64)	0.002	24.49 (22.51–26.47) ^b^	41.64 (40.51–42.78)	<0.0001
Ethnic minorities	18.65 (13.15–24.15) ^b^	10.42 (8.5–12.33) ^b^	0.001	46.11 (39.08–53.15)	45.25 (42.13–48.37)	0.826	35.23 (28.49–41.98) ^a^	44.33 (41.22–47.44)	0.020
Gestational age									
First trimester	26.62 (22.59–30.66)	11.48 (10.19–12.77)	<0.0001	51.52 (46.95–56.08)	45.96 (43.95–47.98)	0.029	21.86 (18.09–25.63)	42.56 (40.56–44.56)	<0.0001
Second trimester	26.29 (23.43–29.16)	13.11 (11.98–14.24)	<0.0001	47.19 (43.95–50.44)	45.56 (43.89–47.23)	0.380	26.51 (23.64–29.38)	41.33 (39.68–42.98)	<0.0001
Third trimester	22.52 (19.27–25.77)	12.87 (11.54–14.2)	<0.0001	50.71 (46.82–54.6)	44.84 (42.86–46.82)	0.008	26.77 (23.32–30.22)	42.29 (40.32–44.25)	<0.0001
Pregnancy frequency									
1	23.00 (20.70–25.31) ^a^	11.69 (10.66–12.72) ^a^	<0.0001	49.69 (46.94–52.43)	44.48 (42.88–46.07)	0.001	27.31 (24.86–29.75)	43.84 (42.24–45.43) ^a^	<0.0001
2	28.57 (24.85–32.29) ^b^	12.46 (11.3–13.61) ^a^	<0.0001	48.50 (44.38–52.62)	46.28 (44.53–48.03)	0.329	22.93 (19.46–26.39)	41.26 (39.54–42.99) ^a,b^	<0.0001
3+	30.43 (23.32–37.55) ^b^	15.34 (13.4–17.28) ^b^	<0.0001	49.07 (41.34–56.8)	46.32 (43.64–49)	0.509	20.50 (14.26–26.74)	38.35 (35.73–40.96) ^b^	<0.0001
BMI before pregnancy									
Thin	43.88 (34.04–53.71) ^a^	18.69 (14.31–23.06) ^a^	<0.001	42.86 (33.05–52.66)	46.23 (40.63–51.83)	0.373	13.27 (6.54–19.99) ^a^	35.08 (29.73–40.44) ^a^	<0.0001
Normal	26.52 (23.69–29.34) ^b^	13.32 (12.24–14.39) ^b^	<0.0001	47.39 (44.19–50.59)	46.71 (45.13–48.29)	0.950	26.09 (23.28–28.9) ^b^	39.97 (38.42–41.53) ^a^	<0.0001
Overweight	21.50 (18.47–24.53) ^b^	11.63 (10.42–12.84) ^b^	<0.0001	51.91 (48.22–55.6)	44.24 (42.37–46.12)	<0.001	26.59 (23.33–29.85) ^b^	44.13 (42.25–46.01) ^b^	<0.0001
Obesity	23.28 (18.16–28.4) ^b^	11.00 (9.34–12.65) ^b^	<0.0001	51.53 (45.47–57.58)	44.21 (41.58–46.84)	0.029	25.19 (19.93–30.45) ^b^	44.79 (42.16–47.42) ^b^	<0.0001
Season									
Spring	/	9.60 (7.67–11.53) ^c^	/	21.82 (10.89–32.74) ^b^	40.51 (37.3–43.73) ^b^	0.006	78.18 (67.26–89.11) ^a^	49.89 (46.61–53.16) ^a^	<0.0001
Summer	36.00 (22.68–49.32) ^a^	30.00 (13.60–46.4) ^a^	0.584	50.00 (36.13–63.87) ^a^	66.67 (49.79–83.54) ^a^	0.150	14.00 (4.37–23.63) ^c^	3.33 (0–9.76) ^c^	0.128
Autumn	30.95 (28.02–33.87) ^a^	16.68 (14.99–18.37) ^b^	<0.0001	49.64 (46.48–52.8) ^a^	55.62 (53.36–57.87) ^a^	0.003	19.42 (16.92–21.92) ^c^	27.70 (25.67–29.73) ^b^	<0.0001
Winter	20.15 (17.58–22.72) ^b^	11.55 (10.69–12.40) ^b^	<0.0001	50.53 (47.33–53.74) ^a^	42.65 (41.33–43.97) ^b^	<0.0001	29.32 (26.4–32.23) ^b^	45.8 (44.47–47.13) ^a^	<0.0001
Vitamin D supplement intake									
No	24.82 (22.82–26.82)	11.74 (11.05–12.44) ^b^	<0.0001	49.30 (46.99–51.62)	43.24 (42.17–44.32)	0.002	25.88 (23.85–27.91)	40.54 (39.47–41.6) ^a^	<0.0001
Yes	28.89 (22.26–35.52)	18.53 (14.55–22.5) ^a^	0.006	51.67 (44.36–58.97)	49.59 (44.47–54.71)	0.649	19.44 (13.66–25.23)	31.88 (27.11–36.65) ^b^	0.002
unknown	24.24 (9.61–38.88)	/		36.36 (19.94–52.79)	/		39.39 (22.71–56.08)	/	
Education									
Primary	29.66 (22.21–37.10)	10.71 (8.09–13.34)	<0.0001	42.76 (34.70–50.82)	42.48 (38.28–46.68)	0.952	27.59 (20.31–34.87)	46.80 (42.56–51.05) ^a^	<0.0001
Medium	24.98 (22.55–27.41)	12.99 (11.97–14)	<0.0001	50.94 (48.14–53.75)	46.62 (45.12–48.12)	0.008	24.08 (21.68–26.48)	40.40 (38.92–41.87) ^b^	<0.0001
Advanced	24.53 (21.19–27.87)	12.35 (11.25–13.45)	<0.0001	47.66 (43.78–51.53)	44.50 (42.83–46.16)	0.140	27.81 (24.34–31.29)	43.15 (41.49–44.81) ^a^	<0.0001
Marital status									
Married	25.46 (23.53–27.38) ^a^	12.53 (11.81–13.25)	<0.0001	49.14 (46.93–51.35)	45.41 (44.33–46.49)	0.003	25.41 (23.48–27.33)	42.06 (40.99–43.14) ^a^	<0.0001
Unmarried	10.53 (0.76–20.29) ^b^	17.91 (8.73–27.09)	0.314	57.89 (42.18–73.61)	52.24 (40.28–64.20)	0.577	31.58 (16.79–46.37)	29.85 (18.89–40.81) ^b^	0.853
Annual family income per capita									
Low	22.86 (19.58–26.14)	9.11 (7.76–10.47) ^b^	<0.0001	50.79 (46.89–54.70)	43.08 (40.75–45.41) ^b^	<0.001	26.35 (22.91–29.79)	47.81 (45.46–50.16) ^a^	<0.0001
Mid	24.73 (21.63–27.83)	11.45 (9.92–12.97) ^b^	<0.0001	50.94 (47.35–54.54)	42.81 (40.45–45.18) ^b^	<0.001	24.33 (21.24–27.41)	45.74 (43.35–48.12) ^a^	<0.0001
High	28.57 (24.45–32.69)	14.39 (12.81–15.97) ^a^	<0.0001	45.24 (40.70–49.78)	48.89 (46.63–51.14) ^a^	0.159	26.19 (22.18–30.20)	36.72 (34.55–38.89) ^b^	<0.0001
NR	26.47 (19.83–33.11)	14.11 (12.84–15.38) ^a^	<0.0001	47.65 (40.13–55.16)	46.19 (44.37–48.00) ^a,b^	0.711	25.88 (19.29–32.47)	39.70 (37.92–41.48) ^b^	<0.001

^a,b,c^ refers to statistical difference in subgroups, *p* < 0.05.

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
