# Peer review of "Vitamin D Nutritional Status of Chinese Pregnant Women, Comparing the Chinese National Nutrition Surveillance (CNHS) 2015–2017 with CNHS 2010–2012"

_nutrients, 2021, doi:10.3390/nu13072237_

Round 1

Reviewer 1 Report

Thanks for the manuscript, it is a good report with solid data to support the conclusions. Here are several points to be considered.

  1. In the Method section, please describe if any inclusion and/or exclusion criteria were used in the study.
  2. Lines 75-77 indicated that the number of subjects was 2250 for CHNS 2010-2012, and 9000 for CHNS 2015-2017. But in Table 1, the number of subjects is 2006 and 8200, respectively. Please explain the difference.
  3. Lines 95-87 and 89-91 described how the location was defined. One method was based on Latitude (tropical, subtropical, warm temperate and medium temperate zones), another approach was according to geographic landmarks (northern and southern). Is there an overlap between these 2 approaches? What is the advantage to apply these 2 methods? How about use one of them?
  4. In lines 97-98, “annual family income was divided into low, middle and 97 high levels according to the three digit method”. Please clarify the criteria of income levels.
  5. In Table 1 under “BMI before pregnancy”, why the sum of percentages of different body weight status for CHNS 2010-2012 is over 100%?
  6. Figure 1 and table 2 showed the same information, suggest to keep table 2 and remove figure 1.
  7. Please revise Figure 2, the graph is not corresponding to the OR values.
  8. In Figure 2, location based on Latitude (tropical, subtropical, warm temperate and medium temperate zones), and geographic landmarks (northern and southern) are among the risk factors. Has the potential correlation of these 2 variables been considered in the multivariable logistic regression?
  9. In the Discussion section, please highlight the novelty of the findings in the present study.

Author Response

Response to Reviewer 1 Comments

Point 1: In the Method section, please describe if any inclusion and/or exclusion criteria were used in the study.

Response 1: All the surveillance sites in CNHS were selected based on all the monitoring sites of Disease Surveillance points system from 31 provincial-level administrative divisions (PLADs) in the mainland of China using stratified, multistage, and random sampling design. People with serious physical and mental diseases were excluded from the surveillance.

30 pregnant women were selected in each surveillance point covering first, second and third trimester. In CHNS 2010-2012, there were a total of 150 monitoring points with 30 pregnant women in each surveillance site, so there should be 4,500 pregnant women samples. However, the vitamin D nutritional status in CNHS 2010-2012 was not a formal indicator, and it was an exploratory indicator of our research group. The blood samples were given priority for the detection of official indicators. Due to the amount of blood samples of pregnant women, it is not possible to perform vitamin D detection for all the samples. Therefore, we used simple random sampling to draw 50% of the samples (that is, 2250) to understand the vitamin D nutritional status of pregnant women. A total of 2006 samples of pregnant women were included in CNHS 2010-2012 after excluding blood samples with hemolysis or incomplete basic information. In CNHS 2015-2017, all the serum samples (9060) of pregnant women were detected.

We added the exclusion criteria in Part 2.1 Subjects and ethics. We also added above information and two references (Ref 21,22) of the sampling methods in both rounds of surveillance in the revised manuscript.

Point 2: Lines 75-77 indicated that the number of subjects was 2250 for CHNS 2010-2012, and 9000 for CHNS 2015-2017. But in Table 1, the number of subjects is 2006 and 8200, respectively. Please explain the difference.

Response 2: In this study, the concentration of serum 25(OH)D in 2250 serum samples of pregnant women from CNHS 2010-2012 and 9,060 serum samples from pregnant women of CNHS 2015-2017 were tested. There are a small number of serum samples were hemolysis or with insufficient serum volume. And the related information of some pregnant women in the basic information database was incomplete. Finally, 2006 and 8200 pregnant women were included in the study after excluding those with unqualified blood samples and/or basic information. We described above information in 2.1 Subjects and ethics section and the first paragraph in 3.1 Serum 25 hydroxyvitamin D concentration in CNHS 2015-2017 and CNHS 2010-2012 section.

Point 3: Lines 95-87 and 89-91 described how the location was defined. One method was based on Latitude (tropical, subtropical, warm temperate and medium temperate zones), another approach was according to geographic landmarks (northern and southern). Is there an overlap between these 2 approaches? What is the advantage to apply these 2 methods? How about use one of them?

Response 3: Thank you so much for your suggestion. We admit the two methods are overlapped. We consider that the latitude division can better show the impact of different UVB radiation levels on vitamin D nutritional status. On the other hand, the mainland of China was usually divided into the South and the North in many published papers. In order to facilitate future related studies to compare vitamin D levels and deficiency rates in pregnant women, we retained the South and North in Table 1 and table 2. In the logistics regression, the variable Location was deleted in the revised version. And we added the description in the part 2.4 Data analyses.

Point 4: In lines 97-98, “annual family income was divided into low, middle and 97 high levels according to the three digit method”. Please clarify the criteria of income levels.

Response 4: Considering the changes in the income level of the two rounds of CNHS, we use the three-digit method to divide the income level and we added the cutoffs in Part 2.2 Data collection.

Point 5: In Table 1 under “BMI before pregnancy”, why the sum of percentages of different body weight status for CHNS 2010-2012 is over 100%?

Response 5:We are sorry about the mistake! We have revised the proportion in Table 1 under “BMI before pregnancy”. We also checked the proportions under other variables in Table 1.

Point 6: Figure 1 and table 2 showed the same information, suggest to keep table 2 and remove figure 1.

Response 6: Thank you for your suggestion! We have removed Figure 1 in the revised manuscript.

Point 7: Please revise Figure 2, the graph is not corresponding to the OR values.

Response 7: We have revised the graph in the revised manuscript.

Point 8: In Figure 2, location based on Latitude (tropical, subtropical, warm temperate and medium temperate zones), and geographic landmarks (northern and southern) are among the risk factors. Has the potential correlation of these 2 variables been considered in the multivariable logistic regression?

Response 8: According to your suggestion, we deleted the Location variable, reanalyzed the logistics regression analysis, updated the OR values in the figure and the corresponding text, and revised the figure.

Point 9: In the Discussion section, please highlight the novelty of the findings in the present study.

Response 9: We added some descriptions about the novelty of the findings in the discussion section. It is the latest report about the vitamin D status of pregnant women in China with national representative. And we adopted the same method and reagents to detect the 25(OH)D concentration. On this basis, we conducted the comparison of vitamin D status in two rounds of surveys among pregnant women for the first time.

Reviewer 2 Report

Manuscript ID: nutrients-1270939

Title: Vitamin D nutritional status of Chinese pregnant women, 2010-2017.

The aim of this study was to evaluate the vitamin D (vitD) status of pregnant women in the latest CNHS (China Nutrition and Health Surveillance) 2015-2017, and analyze the risk factors for vitD deficiency (VDD). Moreover, the authors also compare the vitD status and risk factors of pregnant women in CNHS 2015-2017 with that in CNHS 2010-2012.

Comments and Suggestions for Authors:

The manuscript is a very interesting study, but requires some considerations.

The manuscript pages are incorrectly numbered and must be renumbered from page 6 onwards.

Page 1, Line 2: Title. “Vitamin D nutritional status of Chinese pregnant women, 2010-2017”. The phrase would give more information specifying: comparing the CNHS 2015-2017 with CNHS 2010-2012.

Page 1, Line 13. Abstract. “Optimal vitamin D (vitD) status is beneficial for both pregnant women and their preterm newborns”. Optimal vitamin D (vitD) status is beneficial for both pregnant women and their newborns, and not just for the preterms. The word preterm should be excluded.

Page 1, Line 29. Keywords. There is a mistake in the word: 25-hydroxyviramin D. It should be: 25-hydroxyvitamin D.

Tables 1 and 2. Where it says "Agegroup" should separate: Age group.

Page 6, Line 215. “This may be related to changes in multiple factors such as lifestyle, environment and etc. The change of vitD insufficiency during the two rounds of survey was not as much as that of deficiency rate and sufficiency rate”. This paragraph should not be in the Results section. It should be moved and discussed in the Discussion section.

Page 10, Line 227. "In 2010-2012, the result of univariable logistic regression analysis showed that the vitD inadequacy (including insufficiency and deficiency, 25 (OH) D <12ng/mL)". In this paragraph the considered level of insufficiency should be shown or, given that it has already been indicated previously, remove the deficiency level.

Page 12, Line 349. The authors honestly acknowledge some limitations of the study. However, others should be considered and included.

Demographic data (including age, nationality), education, pregnancy frequency, gestation age, marital status, use of vitD supplementation was recorded based on self-report, which may involve reminder or wishful thinking bias.

All blood samples should have been taken uniformly at the different stations considered. However, only 0.37% blood samples were taken from summer in the CNHS 2015-2017 and only 5.23% blood samples were taken from spring and summer in the CNHS 2010-2012. This should also be stated as a limitation of the study.

In the Materials and Methods section, it is not described whether the blood sample was taken at a certain point in the pregnancy. If not, there may be a bias that those who do it at the end may have had less time for vitD supplements to be recommended or to take effect than those who do it early in pregnancy. This should be taken into account and discussed.

Author Response

Response to Reviewer 2 Comments

Point 1: The manuscript pages are incorrectly numbered and must be renumbered from page 6 onwards.

Response 1: We have renumbered the pages in the revised manuscript.

Point 2: Page 1, Line 2: Title. “Vitamin D nutritional status of Chinese pregnant women, 2010-2017”. The phrase would give more information specifying: comparing the CNHS 2015-2017 with CNHS 2010-2012.

Response 2: We have revised the title according to your kind suggestion.

Point 3: Page 1, Line 13. Abstract. “Optimal vitamin D (vitD) status is beneficial for both pregnant women and their preterm newborns”. Optimal vitamin D (vitD) status is beneficial for both pregnant women and their newborns, and not just for the preterms. The word preterm should be excluded.

Response 3: We have deleted the word preterm in the revised manuscript.

Point 4: Page 1, Line 29. Keywords. There is a mistake in the word: 25-hydroxyviramin D. It should be: 25-hydroxyvitamin D.

Response 4: We are sorry about the typing error, and we have revised it in the keywords section.

Point 5: Tables 1 and 2. Where it says "Agegroup" should separate: Age group.

Response 5: We revised the expression in the revised version.

Point 6: Page 6, Line 215. “This may be related to changes in multiple factors such as lifestyle, environment and etc. The change of vitD insufficiency during the two rounds of survey was not as much as that of deficiency rate and sufficiency rate”. This paragraph should not be in the Results section. It should be moved and discussed in the Discussion section.

Response 6: We have moved the sentence “This may be related to changes in multiple factors such as lifestyle, environment and etc.” to the discussion section.

Point 7: Page 10, Line 227. "In 2010-2012, the result of univariable logistic regression analysis showed that the vitD inadequacy (including insufficiency and deficiency, 25 (OH) D <12ng/mL)". In this paragraph the considered level of insufficiency should be shown or, given that it has already been indicated previously, remove the deficiency level.

Response 7: It was a mistake in the previous version. The cutoff should be 25 (OH) D <20ng/mL. Since we have already indicated previously in the 2.3 section, we removed the deficiency level according to your kind advice.

Point 8: Page 12, Line 349. The authors honestly acknowledge some limitations of the study. However, others should be considered and included.

Demographic data (including age, nationality), education, pregnancy frequency, gestation age, marital status, use of vitD supplementation was recorded based on self-report, which may involve reminder or wishful thinking bias.

Response 8: In the field work of CNHS, all participants need to show their identity cards, so the nationality and age information can’t be wrong, but other self-reported information may involve reminder or wishful thinking bias. We added the limitation in the revised version.

Point 9: All blood samples should have been taken uniformly at the different stations considered. However, only 0.37% blood samples were taken from summer in the CNHS 2015-2017 and only 5.23% blood samples were taken from spring and summer in the CNHS 2010-2012. This should also be stated as a limitation of the study.

Response 9:Due to the actual situation of the implementation of the field work, the seasonal distribution of sample size is indeed the limitation of this study. We added it in the limitation paragraph in Discussion section.

Point 10: In the Materials and Methods section, it is not described whether the blood sample was taken at a certain point in the pregnancy. If not, there may be a bias that those who do it at the end may have had less time for vitD supplements to be recommended or to take effect than those who do it early in pregnancy. This should be taken into account and discussed.

Response 10: During the sampling, pregnant women in different gestational status were selected, and their blood was collected only once on the day of the survey. We analyzed the 25(OH)D concentration in different periods of pregnancy. The results showed that the 25(OH)D concentration of those who took vitD supplements in different gestational periods was higher than those did not take vitD supplements. There was no statistical difference between three gestational periods. We added above information and discussion in the revised version. We understand that it is difficult to unveil the relationship between vitD supplements, time and effect in a cross-sectional study. Therefore, this is a very meaningful topic in the future study.